# Solving hyperbolic conservation laws with characteristic based neural network

## Abstract

Neural network PDE solvers have recently gained popularity.However, it faces difficulty to deal with sharp discontinuity like shock waves in hyperbolic conservation laws.In this paper we propose a characteristic-based neural network to solve one dimension hyperbolic laws.The smooth solution can be derived by equation of characteristic lines ,and shock waves are decided by simple ODE solver.This method achieves a high accuracy with high efficiency. In the future it is hopeful to apply this method to higher dimension problems.

## 1 Introduction

Hyperbolic conservation law is one of the important equations that represent fluid conservation law . It is widely used in fluid mechanics, electromagnetic field theory and other fields,including convection equation, Burger's equation, Euler equations, etc. For nonlinear hyperbolic conservation law equations, even smooth the initial value could result in complex local discontinuities . Therefore, more special numerical methods are required to obtain high-precision approximate solutions.

Neural networks have been widely utilized as PDE solver due to its excellent capacity of approximation . One popular example is physics-informed neural network(PINN)Raissi et al. (2019) , which approximates the PDE with neural network and utilizes auto-differentiation to construct loss function.The result of PINN does not guarantee physical nature ,therefore more targeted methods incorporating physics into problem formulating are developed.MüllerMüller (2023) constructs a neural network with built-in symmetry that preserves the Lagrangian exactly.The prediction of trajectories is more reliable due to the inherent conservation of network.Other approaches attempt to combine machine learning with traditional numerical methods like finite point methodMorand et al. (2024)Chen et al. (2024), discontinuous Galerkin methodArora (2023),WENO Wang et al. (2020).

In fact, due to the discontinuities in hyperbolic conservation systems , it is challenging to directly apply neural network to approximate solution,which will lead to paradoxical results and considerable error. To tackle this issue, some novel methods have been developed.Conservative physics-informed neural network (cPINN) Jagtap et al. (2020)divides the original domain into small subdomains and apply PINN on each of them.To obtain conservation , constraints of flux on the interface of subdomains are added to the overall loss function .Compared to PINN, cPINN performs better on complex-shaped domains and equations with discontinuous solutions.Li et al.Liu et al. (2023) proposed PINNs-WE(Physics-Informed Neural Networks with Equation Weight) to identify the location of shock waves and adjust the loss function dynamically during the training. The Rankine–Hugoniot (RH) relation is also incorporated to obtain conservation at discontinuous region. NDNN(Non-Diffusive Neural Network)Lorin & Novruzi (2024) represent the position of discontinuity lines by neural networks and decompose the global space-time region accordingly. This method is capable of calculating complex phenomena like shock wave interaction and shock wave generation in one dimensional hyperbolic conservation law.

The researches mentioned above adopt different approaches of establishing neural networks to tackle the discontinuities , but all of them require formulation of complex loss functions (NDNN, cPINN) or intricate tuning of parameters Liu et al. (2023) ,and the convergence is hard to guarantee under complex circumstances like multiple shock waves.In fact, the features of hyperbolic conservation law have not been fully explored .

In this paper we propose a novel characteristic-based neural network(CBNN) to solve hyperbolic conservation laws.In this method,the space-time region is divided into subdomains each influenced by a part of the initial value function , and the solution on each subdomain is easily obtained using characteristic lines.The shock wave is determined afterwards by solving the ODE of R-H relation.The training process of each process is independent and convenient,and the shock wave is easily derived by ode solver like Euler method. This method is robust and efficient dealing with diverse type of discontinuities.

In section 2 we will introduce some preliminaries of one-dimensional hyperbolic conservation laws,and apply CBNN to different type of discontinuities like shock wave interaction and shock wave generation.Section 3 gives a theoretical proof of convergence and some numerical experiments are conducted in section 4.Finally a brief conclusion is in section 5.

## 2 METHOD

### 2.1 PROBLEM SET UP

we consider the one-dimensional hyperbolic conservation law which has form as follows:

$$\begin{cases} u_t + f(u)_x = 0, x \in \Omega, t \in [0, T] \\ u(x, 0) = u_0(x), x \in \Omega \end{cases} \tag{1}$$

. This study assumes that $f$ and the solution $u$ is differentiable a.e. ,so the equation can be rewritten as

$$\begin{cases} u_t + \lambda(u)u_x = 0, x \in \Omega, t \in [0, T] \\ u(x, 0) = u_0(x), x \in \Omega \end{cases} \tag{2}$$

where $\lambda = f'$ denotes the derivative of $f$. One important feature of this problem is known as the characteristic line.On the curve

$$\frac{dx}{dt} = \lambda(u), \tag{3}$$

the solution $u$ remains a constant.If in a region $\Omega$ none of the characteristic lines intersect, the solution in $\Omega$ can be expressed as

$$u = u_0(x - t\lambda(u)). \tag{4}$$

we use fully connected neural network $u_\theta(x, t)$ to approximate each smooth region of the solution ,where $\theta$ denotes the parameters of the neural network.Our goal is to approximate the solution on each subdomain separately by characteristic lines and calculate the position of shock waves afterwards, thus deriving a correct weak solution.

### 2.2 CLASSICAL SOLUTION

Let $\Omega = (a, b)$ and $\Gamma$ be the range of influence of $\Omega$.If the function $\lambda(u_0(x))$ is continuous and non-decreasing on $\Omega$, the characteristics of the equation would not cross in the region, so there exists a unique classical solution.According to the method of characteristics, we can construct the equation as follows:

$$u(x, t) = u_0(x - \lambda(u(x, t))t) \tag{5}$$

. Given initial value $u_0$ on $\Omega$, a characteristics-based neural network is obtained to approximate the smooth solution on region $\Gamma$, the parameters $\theta$ will be derived by minimizing the following loss function:

$$\mathcal{L}(\theta) = \frac{1}{N_b} \sum_{i=1}^{N_b} |u_\theta(x_b^i, t_b^i) - u_0(x_b^i - \lambda(u_\theta(x_b^i, t_b^i))t_b^i)|^2 \tag{6}$$

Where $x_b^i(i = 1, 2, \cdots, N_b)$ are training points sampled from the region of influence of $\Omega$, which can be expressed as

$$T_\Omega = \{(x, t) | x \in [a + \lambda(u_0(a))t, b + \lambda(u_0(b))t], t \in [0, T]\} \tag{7}$$

Therefore, $u_\theta(x, t)$ is a approximation of the solution propagated from initial region $\Omega$.

## 2.3 RAREFACTION WAVE

If the function $\lambda(u_0(x))$ is not continuous at some point $x_0$ and $\lambda(u_0(x_0^-)) < \lambda(u_0(x_0^+))$,a rarefaction wave is needed to connect the two adjacent classical solution.The rarefaction wave satisfying entropy condition can be written as:

$$u(x,t) = \lambda^{-1}(\frac{x-x_0}{t}) \tag{8}$$

Rarefaction wave and its adjacent classical solutions are continuous , so in the following analysis, rarefaction wave and classical solution will be treated as smooth solution altogether.

## 2.4 ONE SHOCK WAVE

Shock wave starting at $t = 0$ is caused by discontinuity of initial value $u_0(x)$. Let $x = x_0$ be a discontinuity point of $u_0(x)$ ,if $\lambda(u(x_0^-)) > \lambda(u(x_0^+))$,a shock wave emerges at $x = x_0$ and propagates following the R-H condition.Denote the position of shock wave at time $t$ by $s(t)$, $s(t)$ satisfies the following ODE:

$$\frac{ds(t)}{dt} = \frac{f(u(s(t)^-,t) - f(u(s(t)^+,t)}{u(s(t)^-,t) - u(s(t)^+,t)} \tag{9}$$

Assume $\Omega_1 = (a,x_0), \Omega_2 = (x_0,b)$ are two adjacent regions and a shock wave starts at $x = x_0$. According to 3.1 and 3.2, we can separately derive the smooth solution propagated from $\Omega_1$ and $\Omega_2$,denoted by $u_1$and $u_2$. $T_{\Omega_1}$ and $T_{\Omega_2}$overlap and both contain the curve of shock wave.The position of shock wave could be derived by forward Euler method:

$$\frac{s(t_{n+1}) - s(t_n)}{\Delta t} = \frac{f(u_1(s(t_n),t_n)) - f(u_2(s(t_n),t_n))}{u_1(s(t_n),t_n) - u_2(s(t_n),t_n)} \tag{10}$$

Then the solution could be revised as

$$u(x,t) = \begin{cases} u_1(x,t), x < s(t), (x,t) \in \Omega \times [0,T] \\ u_2(x,t), x > s(t), (x,t) \in \Omega \times [0,T] \end{cases} \tag{11}$$

## 2.5 SHOCK WAVE INTERACTION

Shock waves may intersect and combine into a new shock wave. Assume shock waves $s_1(t)$ separates smooth solution $u_1(x,t)$ and $u_2(x,t)$ ,shock wave $s_2(t)$ separates $u_2(x,t)$ and $u_3(x,t)$,$s_1(t)$ and $s_2(t)$ intersect between time stamp $t_k$ and $t_{k+1}$,in other words,

$$s_1(t) < s_2(t), t = t_k$$
$$s_1(t) \geq s_2(t), t = t_{k+1}$$

Using linear interpolation, the approximated position and time of shock wave interaction are

$$x^* = \frac{s_2(t_k)s_2(t_{k+1}) - s_2(t_k)^2 - s_1(t_k)s_1(t_{k+1}) + s_1(t_k)^2}{(s_1(t_{k+1}) - s_1(t_k)) - (s_2(t_{k+1}) - s_2(t_k))}$$
$$t^* = \frac{t_{k+1}(s_1(t_k) - s_2(t_k)) + t_k(s_2(t_{k+1}) - s_1(t_{k+1}))}{s_1(t_k) - s_2(t_k) + s_2(t_{k+1}) - s_1(t_{k+1})} \tag{12}$$

The generated new shock $s_{new}(t)$ wave starting at $(x^*,t^*)$ follows the R-H rule:

$$\frac{ds_{new}(t)}{dt} = \frac{f(u_1(s(t_n),t_n)) - f(u_3(s(t_n),t_n))}{u_1(s(t_n),t_n) - u_3(s(t_n),t_n)} \tag{13}$$

The same method in 3.3 could be applied.

## 2.6 SHOCK WAVE GENERATION

On $\Omega = (a,b)$ where $\min_{x\in\Omega}\lambda'(u_0)u_0'(x) < 0$, the shock wave starts at $t^* = -\frac{1}{\min_{x\in\Omega}\lambda'(u_0)u_0'(x)}$ and locates on the characteristic of initial point $x_0 = \arg\min_{x\in\Omega}\lambda'(u_0)u_0'(x)$.

Therefore, after calculating the smooth solution of time $t \in [0,t^*]$, the remaining solution can be solved utilizing the similar method in section 3.3.The initial value is $u_\theta(x,t^*)$,shock wave emerges at location $x = x_0$.

## 3 THEORETICAL RESULT

In this section, we present some analysis for convergence of CBNN. Firstly, we consider the error of approximating classical solution in section 3.1.First we make an assumption as follows:

**Assumption 1.** *On $\Omega = (a, b)$,initial value $u_0$ is monotone and differentiable .*

In fact, by dividing $\Omega$ into subintervals where $u_0$ is monotone, and apply the following theorem, the same conclusion can be derived.

**Lemma 1.** *Assuming the exact solution is $\tilde{u}$, the mean square error of neural network solution $u_\theta$ on sample points can be controlled by loss function :*

$$MSE(u_\theta) = \frac{1}{N_b} \sum_{i=1}^{N_b} |u_\theta(x_b^i, t_b^i) - \tilde{u}(x_b^i, t_b^i)|^2 \leq \mathcal{L}(\theta) \tag{14}$$

*Proof.* Without loss of generality we assume $u_0' \geq 0$, then $\lambda'(x) \geq 0$.At position $(x, t)$,define

$$G_{(x,t)}(u) := u - u_0(x - \lambda(u)t),$$

$\tilde{u}(x_b^i, t_b^i)$ is the unique root of $G_{(x_b^i, t_b^i)}(u)$,by Rolle's theorem,

$$
\begin{aligned}
&|G_{(x_b^i, t_b^i)}(u_\theta(x_b^i, t_b^i))| \\
=&|G_{(x_b^i, t_b^i)}(u_\theta(x_b^i, t_b^i)) - G_{(x_b^i, t_b^i)}(\tilde{u}(x_b^i, t_b^i))| \\
=&|u_\theta - \tilde{u}|(G'_{(x_b^i, t_b^i)}(\xi)) \\
=&|u_\theta - \tilde{u}|(1 + tu_0'(x - \lambda(\xi)t)\lambda'(\xi)) \\
\geq&|u_\theta - \tilde{u}|
\end{aligned}
\tag{15}
$$

Sum up for all sample points $(x_b^i, t_b^i)$,

$$
\begin{aligned}
&MSE(u_\theta) \\
=&\frac{1}{N_b} \sum_{i=1}^{N_b} |u_\theta(x_b^i, t_b^i) - \tilde{u}(x_b^i, t_b^i)|^2 \\
\leq&\frac{1}{N_b} \sum_{i=1}^{N_b} |G_{(x_b^i, t_b^i)}(u_\theta(x_b^i, t_b^i)) - G_{(x_b^i, t_b^i)}(\tilde{u}(x_b^i, t_b^i))|^2 \\
=&\mathcal{L}(\theta)
\end{aligned}
\tag{16}
$$

$\square$

Theorem 1 implies a $L^2$ convergence for CBNN for calculating smooth solution.Note that our method is a uniform approximate on the time-space region ,so the error does not accumulate over time.Additionally, as number of shock waves increases, the accuracy and training efficiency are still guaranteed , which is a crucial advantage compared to Lorin & Novruzi (2024).

## 4 NUMERICAL EXPERIMENTS

### 4.1 ONE SHOCK WAVE

we consider Burgers equation $u_t + uu_x = 0$ on $\Omega = [-1, 1] \times [0, 1]$, with initial value

$$u_0(x) = \begin{cases} x, x < 0, \\ -2, x > 0. \end{cases} \tag{17}$$

On the left and right side of shock wave ,the solution are $u_1 = \frac{x}{t+1}$ and $u_2 = -2$ respectively. The weak solution can be written as

$$u(x, t) = \begin{cases} \frac{x}{t+1}, x < 2\sqrt{t+1} - 2(t+1), \\ -2, x > 2\sqrt{t+1} - 2(t+1). \end{cases} \tag{18}$$

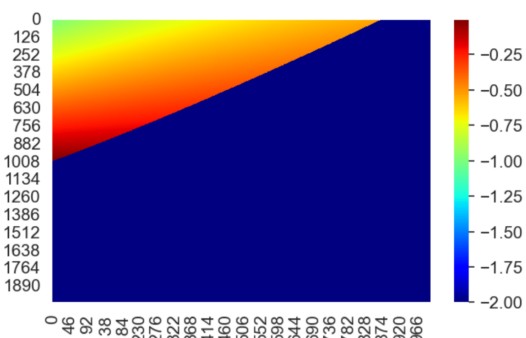

Figure 1: $u_\theta$ for single shock wave,the $L^2$ error is $2.64e-4$

The MSE on sample points is 2.64e-4,the algorithm calculate the position of shock wave success-fully.

For rarefaction wave ,we consider Burger's equation $u_t + uu_x = 0$ on $(-1,1) \times [0,1]$ with discontinuous initial value:

$$u_0(x) = \begin{cases} x, -1, x < 0, \\ 1, 0 < x < \frac{1}{2}, \\ -2, \frac{1}{2} < x < 1. \end{cases} \tag{19}$$

A rarefaction wave emerges at $x = 0$ and connects the smooth solutions on both sides.The shock wave starting from $x = \frac{1}{2}$ encounters the rarefaction wave and smooth solution successively. The shock wave position of (21) is

$$s(t) = \begin{cases} \frac{1}{2} - \frac{1}{2}t, 0 \leq t < \frac{1}{3}, \\ \sqrt{3t} - 2t, \frac{1}{3} \leq t < \frac{3}{4} \\ \sqrt{5(1+t)} - 2(1+t), \frac{3}{4} \leq t \leq 1 \end{cases} \tag{20}$$

The MSE on sample points is 3.12e-4.

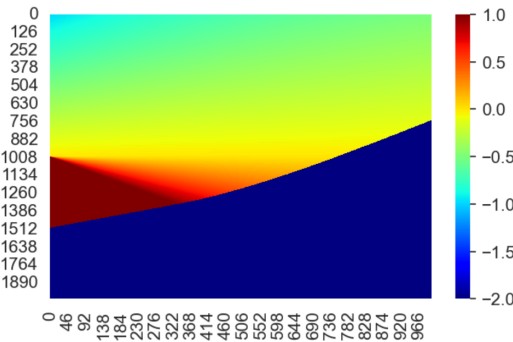

Figure 2: The result for rarefaction wave,the $L^2$ error is 3.12e-4

## 4.2 SHOCK WAVE INTERACTION

As for the case shock waves collide and evolve into a new shock wave, we consider Burger's equation on $(-2,1) \times [0,1]$ with initial value as follows:

$$u_0(x) = \begin{cases} 2, -2 < x < -1, \\ x, -1 < x < 0, \\ -1, 0 < x < 1. \end{cases} \tag{21}$$

Three neural networks approximate two constant regions and a smooth region between them respectively, the total mean square error is 2.28e-4.

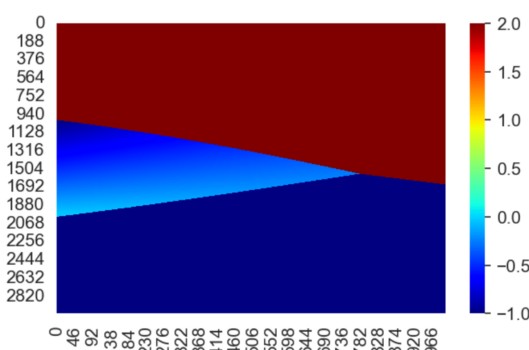

Figure 3: The result for shock wave interaction,the $L^2$ error is 2.28e-4

## 5 CONCLUSION

In this paper, we propose a new method to solve one dimension hyperbolic conservation laws.This characteristic-based neural network applies the rule of characteristic lines and guarantees convergence of solution directly.Complex local discontinuities like shock waves can be derived by R-H relation.This method will be applied to higher dimension and hyperbolic systems in the future.

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
