# OpenReview forum: "Solving hyperbolic conservation laws with characteristic based neural network"
_ICLR.cc/2025/Conference — ICLR 2025 Conference Withdrawn Submission_

### Official Review · Reviewer_7BRc · 2024-10-28

**Soundness:** 2
**Presentation:** 2
**Contribution:** 3
**Rating:** 3
**Confidence:** 3

**Summary:**

This paper introduces a method to solve one-dimensional PDEs modelling hyperbolic conservation laws. The approach aims to incorporate various features specific to these types of equations such as rarefaction waves, shock wave generation and shock wave interactions. The paper includes a theoretical bound on the loss function and carries out numerical experiments for Burger’s equation.

The main idea of the method, as formulated in Section 2.2, is to approximate the PDE solution by a neural network and train it, so as to minimize the loss function formulated in (6). The loss function in (6) is motivated by the representation (5) of the solution. This is similar to the case of PINNs, but instead of aiming to force the neural network to satisfy the PDE, here the approach aims to directly force the neural network to satisfy (5).

**Strengths:**

The idea of formulating a loss function based on the method of characteristics appears to be new. The use of the employed loss function is justified by Lemma 1, which shows that, at least on the training data, the neural network approximation error is bounded by the loss. The numerical results cover different types of discontinuous initial conditions: encompassing an example with a single shockwave, rarefaction waves and shockwave interaction.

**Weaknesses:**

This paper should be rejected, because (1) the paper fails to motivate convincingly the necessity of developing additional neural network-based methods specifically for this class of PDEs, (2) the presentation of the paper makes it difficult to understand how the method incorporates features beyond the classical case, (3) the experiments are limited to a single PDE and no benchmark neural network methods are examined, (4) the paper is very imprecise and unpolished (with wrong punctuation and capitalization throughout), missing many necessary details.

### Main argument
In the case classical case in Section 2.2, the loss function is written out formally.
Beyond this classical case, the authors do not comment on the loss function or the employed neural network, leaving it to the reader to guess how the method extends to the cases of rarefaction waves and shock wave interactions discussed in Section 2.3-2.5. While in principle the developed approach is interesting, other neural network-based methods have already been developed to tackle the same class of PDEs, see Lorin & Novruzi (2024) and Liu et al. (2023) cited in the paper. The paper under review claims that, for these methods, convergence is hard to guarantee under complex circumstances like multiple shock waves. However, no theoretical or experimental evidence is provided and it is also not shown that the new method is capable of addressing this issue. Thus, experiments do not provide convincing evidence of the significance of the proposed approach in comparison to existing methods. Additionally, the new method specifically aims to solve PDEs with discontinuous initial conditions, but Assumption 1 requires that the initial condition is differentiable, hence continuous. Experiments are carried out solely for Burgers' equations and, even though other approaches exist, no benchmark neural network methods are tested. Finally, there are no details regarding the implementation and choices of hyperparameters, see below for a list of questions regarding these.

**Questions:**

### Additional questions to the authors related to the points raised above:

1. Given that you truncate the domain in several parts and solve the PDE separately in each part, it seems that you could also apply PINNs directly on each part - have you considered this approach?
2. You claim that for the approaches by Lorin & Novruzi (2024) and Liu et al. (2023), convergence is hard to guarantee under complex circumstances like multiple shock waves. Can you provide theoretical or experimental evidence for this claim?
3. Did you carry out numerical experiments which show that your approach does not face this issue in presence of multiple shock waves?
4. In the proof of Lemma 1 it is stated that $u_0'\geq 0$ implies $\lambda'\geq 0$. Why is this true?
5. You claim that your theoretical results show that, as the number of shock waves increases, the accuracy and training efficiency are still guaranteed. I could not find any support for this claim, could you be more specific here? The result was only proved under Assumption 1, which seems to exclude shock waves?


### Questions regarding implementation:
The paper leaves open several crucial details related to the implementation. For example:

1. Which neural network architecture was used in experiments?
2. Which training algorithm was used?
3. How many data points were used?
4. How were the exact solutions in the experiments of Section 4.2 computed?
5. What sampling scheme was used to sample the points $x_b^i$?
6. In the experiments, is the reported MSE the minimal neural network loss or the distance to the true solution?
7. On what grid was the $L^2$ error evaluated in the experiments?



### Additional comments that did not impact the score:
1. Capitalizations are missing at the beginning of Sections 2.1 and 4.1
2. Punctuation and spacing is wrong throughout the paper: there should be a space after commas, not before them. There should be a space after each full stop.
3. Please label axes of Figures to allow readers to understand which axes represents $x$ and $t$ variables, respectively.
4. Can you comment on potential limitations for high-dimensional equations? What is the reason that you formulated the method only for one-dimensional PDEs?
5. Once you write Burgers equation, once Burger’s equation.
6. Sections 2.4-2.6 refer to 3.1, 3.2 and 3.3, which does not appear correct.
7. Lemma 1 is referred to as a theorem in several places. Why do you label it a lemma then?

---

### Official Review · Reviewer_J1ZF · 2024-11-01

**Soundness:** 2
**Presentation:** 1
**Contribution:** 2
**Rating:** 3
**Confidence:** 5

**Summary:**

This research paper proposes a novel method for solving hyperbolic conservation laws, a type of equation commonly found in fields like fluid dynamics. The method, called the Characteristic-Based Neural Network (CBNN), leverages the unique properties of hyperbolic conservation laws, specifically the concept of characteristic lines, to simplify the problem. The CBNN divides the problem domain into subregions, each influenced by a distinct portion of the initial conditions. It then approximates the smooth solution within each subregion using the characteristic lines and addresses discontinuities, such as shock waves, by solving ordinary differential equations based on the Rankine–Hugoniot condition. The authors demonstrate the effectiveness of their approach numerical experiments on the Burgers' equation, a well-known example of a hyperbolic conservation law.

**Strengths:**

1. This research propose a method to solve one-dimensional hyperbolic conservation laws by utilizing the method of characteristics and the Rankine-Hugoniot (R-H) condition.
2. The CBNN divides the space-time region into subdomains based on the influence of initial values and approximates solutions on each subdomain independently.
3. This approach allows for accurate computation of solutions, including shock wave interactions and generations.

**Weaknesses:**

1. The paper lacks a comparison with existing methods for solving hyperbolic conservation laws. Benchmarking CBNN against other state-of-the-art techniques is essential to demonstrate its effectiveness and efficiency.
2. The paper presents only a limited number of numerical experiments, primarily focusing on Burgers' equation. To demonstrate the robustness and broader applicability of the proposed method, additional examples involving various hyperbolic conservation laws, as well as more complex initial and boundary conditions, are needed.
3. The paper suffers from numerous typos, grammatical errors, and unclear representation of the concepts. This poor presentation makes it challenging to understand the proposed method and evaluate its merit fully.
4. The paper claims to address the challenges of discontinuities in hyperbolic conservation laws. However, the utilization of the R-H condition for shock wave computation might overlap significantly with the approach presented in NDNN by Lorin and Novruz.  A clear and detailed comparison with NDNN is necessary to distinguish the novelty and contribution of CBNN.

**Questions:**

1. Could the authors provide a comparison of CBNN with existing state-of-the-art methods for solving hyperbolic conservation laws? This would help clarify how CBNN performs relative to established techniques in terms of effectiveness and efficiency.
2. Are there additional numerical experiments that could showcase CBNN's performance on a broader range of hyperbolic conservation laws, with more varied initial and boundary conditions? This would strengthen the demonstration of CBNN's robustness and general applicability.
3. How does the choice of neural network architecture and hyperparameters affect the performance of CBNN?
4. Since CBNN addresses challenges related to discontinuities and the R-H condition is applied for shock wave computation, could you provide a more detailed comparison with the NDNN approach by Lorin and Novruz? This would help to clearly differentiate the novel contributions of CBNN and its unique handling of discontinuities.
5. Could the authors provide an ablation study to assess the contributions of each component within the CBNN framework? This would help in understanding which elements are critical for achieving optimal performance.

---

### Official Review · Reviewer_AjpC · 2024-11-03

**Soundness:** 1
**Presentation:** 1
**Contribution:** 1
**Rating:** 1
**Confidence:** 4

**Summary:**

This paper presents a novel neural network approach for solving one-dimensional hyperbolic conservation laws, focusing on efficiently handling sharp discontinuities, such as shock waves. Traditional neural network solvers for partial differential equations (PDEs) often struggle with accurately capturing these discontinuities. To address this, the authors introduce a characteristic-based neural network that leverages characteristic lines to derive smooth solutions, while using an ordinary differential equation (ODE) solver to resolve shock waves. This method demonstrates high accuracy and computational efficiency. Future work aims to extend this approach to higher-dimensional problems and more complex hyperbolic systems.

**Strengths:**

This paper presents a novel neural network approach for solving one-dimensional hyperbolic conservation laws, focusing on efficiently handling sharp discontinuities, such as shock waves. Traditional neural network solvers for partial differential equations (PDEs) often struggle with accurately capturing these discontinuities. To address this, the authors introduce a characteristic-based neural network that leverages characteristic lines to derive smooth solutions, while using an ordinary differential equation (ODE) solver to resolve shock waves. This method demonstrates high accuracy and computational efficiency. Future work aims to extend this approach to higher-dimensional problems and more complex hyperbolic systems.

**Weaknesses:**

1. Limited to One Dimension: Currently applicable only to one-dimensional problems, limiting its immediate real-world applicability.

2. Challenges in Extending to Higher Dimensions: Potential issues with increased computational complexity and difficulty managing discontinuities in two or three dimensions.

3. Dependence on Characteristic Lines: Heavy reliance on characteristic lines may reduce generalizability for different types of hyperbolic equations or boundary conditions.

**Questions:**

How does the characteristic-based approach handle cases with multiple interacting shocks or rarefaction waves? Have any specific tests been conducted in these scenarios?

What challenges do you foresee in extending this method to two or three-dimensional hyperbolic systems, and how do you plan to address them?

---

### Official Review · Reviewer_mE1d · 2024-11-03

**Soundness:** 1
**Presentation:** 1
**Contribution:** 2
**Rating:** 3
**Confidence:** 4

**Summary:**

This paper proposes a method that combines neural networks with the method of characteristics to solve one-dimensional transport PDEs.

**Strengths:**

The paper proposes a combination of neural networks with a classical numerical method. This is still a poorly explored topic in the literature, with a lot of potential.

The paper has a theoretical component that proposes a practical upper bound on performance.

**Weaknesses:**

The paper is not well written. There are many grammatical mistakes, too many to list. In addition, the math is not clear in some places.

The paper is very sparse and short. There aren't enough numerical experiments.

It is not clear what the role of neural networks is, since so much of the solution is constructed using the method of characteristics,  knowledge about shock formation and interaction, R-H condition, and so on.

The authors seem to misunderstand the definition of "shock". A discontinuity in the initial condition is not a shock. The latter occurs when two or more characteristic lines converge. A simple discontinuity is sometime called a "contact discontinuity".

**Questions:**

Lemma 1 is useful, however, why is $\lambda'(x)>0$? This is necessary for the proof. Can this be assumed without loss of generality?

---

### Note · Authors · 2024-11-17

I have read and agree with the venue's withdrawal policy on behalf of myself and my co-authors.